# Fluoroquinolone-Transition Metal Complexes: A Strategy to Overcome Bacterial Resistance

**DOI:** 10.3390/microorganisms9071506

**Published:** 2021-07-14

**Authors:** Mariana Ferreira, Paula Gameiro

**Affiliations:** REQUIMTE-LAQV (Rede de Química e Tecnologia—Laboratório Associado para a Química Verde), Departamento de Química e Bioquímica, Faculdade de Ciências da Universidade do Porto, Rua do Campo Alegre, s/n, 4169-007 Porto, Portugal; mariana.ferreira@fc.up.pt

**Keywords:** fluoroquinolones, metalloantibiotics, bacterial resistance, liposomes, porins, biophysics, microbiology, fluorescence, surface plasmon ressonance, atomic force microscopy

## Abstract

Fluoroquinolones (FQs) are antibiotics widely used in the clinical practice due to their large spectrum of action against Gram-negative and some Gram-positive bacteria. Nevertheless, the misuse and overuse of these antibiotics has triggered the development of bacterial resistance mechanisms. One of the strategies to circumvent this problem is the complexation of FQs with transition metal ions, known as metalloantibiotics, which can promote different activity and enhanced pharmacological behaviour. Here, we discuss the stability of FQ metalloantibiotics and their possible translocation pathways. The main goal of the present review is to frame the present knowledge on the conjunction of biophysical and biological tools that can help to unravel the antibacterial action of FQ metalloantibiotics. An additional goal is to shed light on the studies that must be accomplished to ensure stability and viability of such metalloantibiotics. Potentiometric, spectroscopic, microscopic, microbiological, and computational techniques are surveyed. Stability and partition constants, interaction with membrane porins and elucidation of their role in the influx, determination of the antimicrobial activity against multidrug-resistant (MDR) clinical isolates, elucidation of the mechanism of action, and toxicity assays are described for FQ metalloantibiotics.

## 1. Introduction

Quinolones, or 4-quinolones, are a family of antibacterial agents commonly used in the clinical practice, synthetized from antimalarial agents [1,2]. Nalidixic acid, the first quinolone, was synthetized in 1962, from a compound isolated from chloroquine. Two years later, it was uncovered that its mechanism of action was related to the inhibition of the bacterial DNA gyrase synthesis [3]. The chemical structure of quinolones is derived from a 4-oxo-1,8-naphthyridine-3-carboxylic acid nucleus (Figure 1), containing a conserved carboxylic acid and an exocyclic oxygen in the positions 3 and 4, and a carbon atom (unsubstituted or substituted) in position 8. The carboxylic acid and exocyclic oxygen are responsible for the antimicrobial activity of this family [1,2,4]. During the last decades, several substitutions in different positions of the nucleus of quinolones have been made to broad and improve their antimicrobial activity. The incorporation of a fluorine atom at the 6th position and a piperazinyl group at the 7th position gave rise to the fluoroquinolones (FQs) family, in the 1970s, granting significant improvements in the antimicrobial activity and potency of the compounds [1,2,5]. Since then, several substitutions in the different positions of the quinolone nucleus have been made, disclosing the structure/activity relationship [6]. Different generations of FQs, classified according to the spectrum of action, potency, pharmacodynamics, and pharmacokinetics (bioavailability) properties, are actually known [1,2]. The mechanism of action of quinolones has been a subject of study, revealing that this family of antibiotics inhibits bacterial replication, interfering with the enzymatic activity of DNA gyrase (topoisomerase II) or topoisomerase IV. These two enzymes play a crucial role in bacterial growth, with DNA gyrase introducing negative supercoils in the DNA strand and topoisomerase IV decatenating the DNA molecules [7,8]. The action of quinolones consists of the formation of a ternary complex with the DNA and the enzymes of bacteria, cleaving the bacterial DNA and hampering bacterial replication [4,9]. Commonly, DNA gyrase is preferentially targeted in Gram-negative bacteria and topoisomerase IV in Gram-positive bacteria [7,8].

However, bacteria have evidenced great ability to overcome the action of antibiotics, through the development of bacterial resistance mechanisms comprising: (i) the reduction of the intracellular concentration of the antibiotic (via efflux pumps, through drug extrusion or reduction of drug influx, by changing the permeability of the membrane or by decreasing the expression of porins) [10,11,12]; (ii) alteration of the drug target (through mutations in genes encoding the target site, by enzymatic alterations of the binding site, and/or by replacement or bypass the target site [11,12]); and/or (iii) antibiotic inactivation by bacterial enzymes [10,11]. The plasticity of bacteria to quickly circumvent the antibiotic action has triggered the need to find/develop new antimicrobial drugs in due time [13,14].

The reported bacterial resistance mechanisms commonly described against FQs occur via modifications of the bacterial topoisomerases (by chromosomal mutations or plasmid-acquired resistance genes), reducing the affinity of the drugs to their main targets, or by reducing the intracellular concentration of the drugs (through extrusion, via efflux pumps, and/or by decreasing drug influx) [7,15]. Concerning the targets of FQs, the susceptibility of the subunits of each topoisomerase to mutations varies among different bacterial strains. The GyrA subunit of DNA gyrase is usually more susceptible in Gram-negative bacteria and ParC subunit of topoisomerase IV in Gram-positive bacteria [3,16]. The residues commonly mutated are enclosed in specific sequences known as “quinolone-resistance-determining region” (QRDR) [11,14,16,17,18]. On the other hand, bacteria can acquire resistance genes via plasmid-mediated quinolone resistance (PMQR). Several genes are commonly described, as *qnrA*, *qnrS*, *qnrB*, *qnrC*, and *qnrD* [14,19,20]. Focusing on the mechanism of reduction of drug intracellular concentration, FQs are commonly extruded from the cytoplasm of bacterial cells through efflux pumps. There are numerous efflux pumps well characterized for different bacterial species, as AcrAB-TolC of *Escherichia coli* (related to genes *acrR* and *marR*), MexAB-OprM, MexCDOpr-J, Mex-EF-OprN, MexXY-OprM, and MexVW-OprM of *Pseudomonas aeruginosa* and MepA, NorA, NorB and NorC of *Staphylococcus aureus* [16,21,22,23]. Commonly, resistant bacteria overexpress the efflux pumps, increasing the minimum inhibitory concentration (MIC) of drugs [24]. Additionally, some efflux pumps, as OqxAB and QepA, are related to PMQR mechanisms [14,16]. Furthermore, the reduction of the influx of FQs is also a common strategy of bacteria to reduce the intracellular concentration of these drugs. Porins are outer membrane proteins (Omps) usually used by FQs to translocate the bacterial membranes of Gram-negative species. The Omps mainly described in the influx of FQs are OmpF, OmpC, OmpE, OmpK, and OmpU [23,25,26,27]. Bacteria have the skills to underexpress, deactivate, and/or switch the type of expressed porins to reduce the influx of these drugs [16,28,29].

One of the strategies to bypass resistance mechanisms to FQs consists of the complexation of these molecules with transition metals, taking advantage of their chemistry, which allows an easily chelation [30,31,32]. In the last decades, several studies focused on the complexation of FQs with transition metals have been performed, evidencing the formation of stable complexes with antimicrobial activity (similar or improved compared to FQs) [30,33,34,35,36] and supporting alternative influx routes [30,37,38,39].

This review is focused on ternary complexes of transition metal ions, 1,10-phenanthroline (phen) and FQs, known as metalloantibiotics, which have been the subject of study due to their promising insights against resistant bacterial strains. The chemical, biophysical, and microbiological characterization of these metalloantibiotics have been extensively performed, including the study of the stability constants, interaction with bacterial model membranes and porins, antimicrobial activity against susceptible and resistant bacterial strains and of the mechanism of action, combining potentiometric, spectroscopic, microbiological, microscopic, and computational approaches, as following presented.

## 2. Fluoroquinolone Metalloantibiotics

The complexation of FQs with transition metal ions (as zinc, copper, cobalt, nickel, palladium, and platinum) have been a strategy widely explored to try to counteract resistant bacteria [30,40,41]. This approach relies on the role of transition metal ions in biological processes [42,43,44,45] and on the chemistry/structure of FQs, which promotes complexation with several metals [30,31,32]. These antibiotics exhibit an amphoteric character due to the presence of the two ionisable groups, the carboxyl group (4th position) and the amino group (of the piperazinyl substituent of the 7th position), respectively (Figure 1). Therefore, each FQ has two specific pK_a_ values, corresponding to the ionisation of the carboxylic acid (pK_a1_) and of the amino group (pK_a2_), usually varying from 5.6–6.4 and 7.6–9.3, respectively. At low pH (pH < pK_a1_), FQs have both groups protonated, existing mainly in the cationic form. With the increasing of the pH, the carboxylic acid deprotonates (pH > pK_a1_), explaining the predominant coexistence of zwitterion and neutral forms at physiological pH [46,47]. At high pH (pH > pK_a2_) the amino group also deprotonates, the anionic form being prevalent in solution [47,48]. The protonation state of the FQs will affect their interaction and ability to penetrate across the membranes [49,50]. Moreover, FQs easily chelate to the transition metals through the groups responsible for their antimicrobial activity, the carboxylic acid and the exocyclic oxygen of the positions 3 and 4 [30,31,32].

During the last decades, numerous studies comprising several FQs and different transition metals have been performed [30,31,33,34,35,51,52,53,54,55,56], including the solution study of binary and ternary complexes. Ternary complexes containing 1,10-phenanthroline (1,10-phen), a nitrogen donor heterocyclic ligand, were extensively explored due to the nuclease activity of this ligand when complexed to copper [57,58,59]. These findings triggered interest in the investigation of copper/phen complexes [60,61] and other complexes containing transition metal ions and phen.

The stability constants of binary—metal(II)/FQ 1:1 and 1:2—and ternary—metal(II)/FQ/phen—complexes, determined by potentiometric titrations, are reported for several FQs (ciprofloxacin–cpx; enrofloxacin-erx; levofloxacin–lvx; moxifloxacin–mxfx; sparfloxacin–spx; norfloxacin–nfx; ofloxacin–ofx; and lomefloxacin-lmx), cobalt(II), nickel(II), zinc(II) and copper(II) [30]. Among all studied complexes, only Cu (II) complexes proved to be stable (Table 1) [30,33,34,35,51]. These results are in agreement with the Irving–Williams series, which indicate that complexes of copper (II) are stronger and more stable than complexes of iron (II), cobalt (II), nickel (II), or zinc (II). However, all the binary complexes revealed dissociation in solution under relevant biological concentrations and physiological pH.

Ternary complexes proved to be more stable than the binary ones, owing to the presence of five- and six-membered ring, the establishment of stacking interactions, and the formation of π-back bonding [30,35,62,63,64], arriving at the crucial contribution from the metal-ligand π-back bonding from the heteroaromatic N-base (i.e., metal d-electrons are pushed into the vacant π * of phen) present in addition to the metal-ligand σ-donation [30]. The speciation analysis also confirmed that only ternary complexes were stable under physiological pH and relevant biological concentrations for microbiological assays. For this reason, subsequent studies were limited to ternary complexes of Cu(II), FQ and phen (CuFQphen). Under physiological conditions, these complexes coexist in a mixture of mono- ([Cu(FQ)phen]^+^) and di-cationic ([Cu(HFQ)phen]^2+^) forms in solution.

The CuFQphen ternary complexes were characterized by elemental analysis, UV-vis, and Fourier-transform infrared spectroscopy (FT-IR) spectroscopies, and by single-crystal X-ray diffraction, revealing that these metalloantibiotics acquire a square pyramidal geometry slightly distorted, lying on a five-coordinated metallic centre [30]. The base of the quadrangular pyramid comprises of two oxygen atoms of the FQ (one of the carboxyl group and other of the exocyclic oxygen) and two nitrogen atoms of the bidentate ligand phen. The axial plane of the pyramid is commonly occupied by a water molecule, coordinated through the oxygen (Figure 2) [30,32,34,65,66]. In the specific case of the metalloantibiotic of cpx, the fifth coordination of the metal occurs through a nitrogen atom of the piperazine ring of another FQ molecule (Figure 3). This specific coordination allows the formation of a polymeric chain composed of Cucpxphen cationic molecules [30].

The structural solidity of the CuFQphen metalloantibiotics is mainly assured by C-H∙∙∙O and weak C-H∙∙∙F hydrogen bonds and π · π stacking interactions between the phen rings of the adjacent complex molecules. The analysis of the synthetised complexes disclosed that counter-ions (typically NO_3_^−^) and crystallisation water molecules also contribute to the stability of these structures, through the establishment of hydrogen bonds (O-H · O and O-H · N) with oxygen atoms of the 3rd and 4th positions of FQs, with nitrogen atoms of the ligands of FQs or with the coordinated water molecules [30,33].

These findings encouraged a deeper characterization of these metalloantibiotics to assess their antibacterial activity, to study the interaction with membranes, to clarify the adopted translocation routes and to elucidate the mechanism of action and the toxicological profile, combining spectroscopic, microscopic, microbiological, and computational techniques, as presented in the following sections.

Among several FQs, metalloantibiotics of cpx, erx, lvx, mxfx, and spx (Figure 4) have been deeply explored.

## 3. Metalloantibiotic-Membrane Interactions

### 3.1. Partition Constants

The cell envelope represents the first natural barrier against antimicrobial drugs. Therefore, the study of the interaction of a drug with membranes and its ability to destabilize and/or translocate this barrier are crucial in the characterization of a new molecule. The chemistry of FQ metalloantibiotics, clearly different from the one of free FQs, suggests different interactions which may support alternative influx routes. One of the main approaches well established to study drug-membrane interactions is the determination of the partition constant ($K_{p}$). The partition of a drug evidences its ability to penetrate into the membrane, being able to diffuse to a deeper region of the membrane or to remain at its surface [67]. Octanol/water was considered a traditional model system for the assessment of drug partition in membranes during decades. Nevertheless, liposomes became excellence model systems to mimic membranes, due to their amphiphilic character (absent in octanol/water systems), anisotropic nature (versus the isotropic nature of octanol) and structural similarity with biological membranes [67,68]. Generally, the partition constants are determined by spectroscopic techniques as nuclear magnetic resonance (NMR) [69,70,71], derivative spectrophotometry [71,72,73,74], and fluorescence spectroscopy [74,75,76,77], the last one being the most sensitive [78]. Additionally, $K_{p}$ values determined by computational approaches have been recently reported [38,79,80,81].

The in-depth study of metalloantibiotics of CuFQphen disclosed numerous $K_{p}$, determined in different membrane model systems of eukaryotes and prokaryotes, as shown in Table 2. Phosphatidylcholine (PC) is the main phospholipidic component of eukaryotic cells [82], 1,2-dimyristoyl-s*n*-glycero−3-phosphocholine (DMPC) and 1-palmitoyl-2-oleoyl-sn-glycero-3-phosphocholine (POPC) being mimetic model systems widely used to mimic these cells. Regarding the prokaryotic cells, phosphatidylglycerol (PG), phosphatidylethanolamine (PE), and cardiolipin (CL) represent the major bacterial phospholipids [83,84]. Therefore, there are distinct mimetic model systems extensively used for bacteria, varying in the mixture complexity, as: 1-palmitoyl-2-oleoyl-s*n*-glycero-3-phospho-(1’-rac-glycerol)–POPG or dimyristoyl-L-α-phosphatidylglycerol (DMPG); binary mixtures (75:25 or 60:40) of 1-palmitoyl-2-oleoyl-s*n*-glycero-3-phosphoethanolamine (POPE) and POPG; a ternary mixture (67:23:10) of POPE:POPG:CL; a ternary mixture (4:5:11) of neutral lipids:CL:PG; and natural bacterial lipidic extracts as the polar and the total *E. coli* lipid extracts [85,86]; among others.

Based on differences of the chemistry of free FQs and metalloantibiotics (neutral/zwitterionic vs. cationic forms, prevalent under physiological conditions, respectively), and taking into account the negative charge characteristic of bacterial membranes (mean zeta potential of the *E. coli* total extract liposomes: −36.8 mV [71]), distinct interactions and potential alternative influx routes in bacteria are expected. Analysing $K_{p}$ values, metalloantibiotics strongly interact with bacterial model systems, showing greater constants compared to FQs [38,71,75,77,87], with no exception. Analysing the $K_{p}$ values obtained by fluorescence and UV-vis spectroscopies, it is possible to conclude that metalloantibiotics with higher percentage of di-cationic form ([Cu(HFQ)phen]^2+^) exhibit higher partition constants: $K_{p}$ Cucpxphen > $K_{p}$ Cumxfxphen ≈ $K_{p}$ Cuspxphen ≈ $K_{p}$ Cuerxphen > $K_{p}$ Culvxphen. The di-cationic form favours the electrostatic interactions with the negative surface of the membranes, while the mono-cationic form ([Cu(FQ)phen]^+^) may act as a pseudo neutral form, as the copper charge is masked by the coordination with phen and FQ. This pseudo neutral form may behave as a lipophilic species, assisting the diffusion into the membrane. Additionally, the molecular dynamics (MD) simulations performed with Cucpxphen revealed low free energy barrier to cross the membrane, favouring the permeation of the metalloantibiotic, and supporting the passive diffusion translocation across the lipid bilayer [38].

Concerning the eukaryotic model systems, metalloantibiotics exhibit reduced $K_{p}$ values (in some cases, there is no partition or cannot be determined, suggesting no interaction), compared to the ones obtained in bacterial models. Moreover, computational experiments performed with Cucpxphen also revealed a reduced affinity of the metalloantibiotic for POPC, when compared to POPG [38]. These results show the selective behaviour of metalloantibiotics to bacterial models, which is positive in terms of antimicrobial therapy.

These insights predict an influx route strongly dependent on hydrophobic interactions [30,38,71,75,77,87], contrarily to what is commonly observed for free FQs (with exception of spx [39,71]), whose translocation occurs through porins and/or by a combination of hydrophobic and hydrophilic routes, via porin/lipid interface [30,39,51,88,89,90]. The distinct translocation routes proposed for free FQs and metalloantibiotics were also corroborated by the changes observed in the emission spectra of the compounds in the presence of increasing amounts of liposomes. Some authors suggest different changes of the dipole moments of FQs and metal antibiotics during the electronic transitions, resultant from distinct interactions between the solvent and the solute, which interfere with the non-radiative and radiative coupling between ground and excited states [75,77,87].

The membrane of Gram-negative bacteria is the main barrier to drug permeation due to its structural complexity. These membranes encompass two asymmetric membranes, differing in their composition. The inner membrane is a phospholipidic bilayer, contrarily to the outer membrane, which comprises of two distinct leaflets. The inner leaflet of the outer membrane is mainly composed by phospholipids [91], while the outer leaflet contains lipopolysaccharides (LPS) [29,91,92,93], lipoproteins, and integral membrane proteins (as porins) [94,95]. This complex membrane environment turns drug permeation into a challenge. For this reason, different techniques concerning the study of the passive diffusion of FQs and metalloantibiotics should be carried out. The in vitro parallel artificial membrane permeability assay (PAMPA), membrane permeabilization assays using carboxyfluorescein and/or SYTOX Green uptake in live bacteria can also be considered as further experiments [96,97,98]. These studies will provide additional information, which will be important to elucidate the contribution of the hydrophilic and hydrophobic components of the influx routes. Moreover, the quantification of the drug uptake can also give important information about the permeation pathways of FQs and metalloantibiotics. Therefore, accumulation studies should be further explored, combining mass-spectrometry-based assays (though liquid chromatography coupled to mass spectrometry) [99] and fluorescence approaches (by microfluidic flows coupled to fluorescence imaging and/or time-lapse fluorescence microscopy) [100,101].

### 3.2. Porins

As previously stated, the influx of free FQs usually depends on outer membrane porins in Gram-negative bacteria. Based on the hypothesis that free FQs and metalloantibiotics adopt distinct influx routes in Gram-negative cells, the role of the porins in the translocation of metalloantibiotics and the porin-metalloantibiotic interaction have been vastly explored. The combination of microbiological, computational, and biophysical (Surface Plasmon Resonance–SPR and fluorescence spectroscopy) approaches have disclosed the role of porins in the transport of the metalloantibiotics [30,35,37,38,39,51]. OmpF and OmpC, the major porins involved in the influx of FQs [25,26], were further explored.

#### 3.2.1. Microbiological Studies

Microbiological assays evaluated the impact of the absence of porins in the MIC of metalloantibiotics, using mutant strains devoid of specific porins, as: JF568, a parental strain derived from *E. coli* K12 and its derived mutants, *E. coli* JF701 (devoid of OmpC) and *E. coli* JF703 (devoid of OmpF) [102,103]; the parental strain *E. coli* W3110 and the mutant strain *E. coli* W3110 ΔFΔC (devoid of OmpC and OmpF) [27,104]; and *E. coli* BE BL21(DE3) and *E. coli* BE BL21(DE3)omp8, derived from the previous one but devoid of the major outer-membrane porins [27]. Analysing the extensive data reported for several metalloantibiotics by Gameiro et al., Sousa et al., Feio et al., and Fernandes et al., it is possible to observe no significant change of the MIC of each metalloantibiotic when the porins are not expressed (comparing parental and mutant strains), evidencing that these molecules can translocate into the bacterial cells in a porin-independent way [30,34,35,51]. Focusing on the analysis of the five metalloantibiotics vastly explored in literature, Cucpxphen and Culvxphen were revealed to be sensitive to the deletion of OmpF, evidencing a pronounced increase of the MIC value against the strain JF703 [30]. Regarding phen, Cu(II)/phen (1:1) and copper solutions, the MIC values reported against all parental strains and respective mutants by Feio et al. were greater than the ones determined for FQs and metalloantibiotics (more than 10^3^-fold), highlighting the complex stability and proving the antimicrobial activity of the metalloantibiotics [30].

The microbiologic data, concerning the clarification of the role of porins for the translocation of metalloantibiotics in bacteria, points out for a permeation pathway independent of porins for most of the complexes, with the exception of Cucpxphen and Culvxphen, which may use OmpF to penetrate the bacterial cell. Nevertheless, Sousa et al. proposed that these results may arise from conformational changes of the bacterial membranes consequent of porin deletion [39].

These results anticipate that metalloantibiotics may be effective against bacterial strains, evidencing decreased permeation through porins. Moreover, combining these outcomes with the partition data, and knowing that membranes of Gram-positive bacteria are absent of porins, it is also expected that metalloantibiotics cross these membranes by passive diffusion, being a good strategy to explore against Gram-positive strains.

#### 3.2.2. Biophysical Studies

Biophysical techniques have also been broadly used to characterize the interactions of drugs as FQs with porins (especially focused on OmpF, the major channel implicated in the transport of FQs in *E. coli* [105]). Fluorescence spectroscopy experiments have been widely reported to determine the association and location of several compounds with these channels [37,38,88,105,106]. More recently, SPR was also validated to study the interaction of the metalloantibiotics with OmpF [37,39]. OmpF is a homotrimeric protein that contains aromatic residues, allowing fluorescent experiments based on its intrinsic fluorescence. Additionally, there are two Trp residues (Trp^214^–W214 and Trp^61^–W61) commonly described in biophysical assays due to their distinct location at the channel (Figure 5), giving information about the phospholipid headgroups region (W214) and the lipid-protein interface (W61) [90,105,106]. The environment surrounding both Trp is hydrophobic, with hydrophobicity being highest surrounding W61 [90,107].

The association ($K_{ass}$) of metalloantibiotics with OmpF, determined in proteoliposomes of OmpF/*E. coli* total extract by SPR and fluorescence spectroscopy was recently reported [37,39]. The $K_{ass}$ values assessed were comparable among the five studied metalloantibiotics and analogous to the ones of free FQs. These results were explained by the sensitivity of the used techniques, which disclose a binding but are not able to detail specific regions of the channel. Moreover, SPR data evidenced a faster association and dissociation of the metalloantibiotics, compared to free FQs [37,39]. Location studies, evaluating the interaction of Cucpxphen, a metalloantibiotic that may use OmpF to penetrate the bacterial cell, with OmpF mutants (lacking W61 and W214), were also performed. The results supported a random location of the metalloantibiotic within the channel [37], contrarily to what was observed for free FQs (as cpx, erx and mxfx), which exhibited a preferential interaction near the centre of the channel (with W61) [90,105]. Additionally, the presence of the metalloantibiotic evidenced implications in the conformation of the channel, which proposes that Cucpxphen may use the porin/bilayer interface for its influx [37].

These studies were also corroborated by computational experiments, extending the studies to Cuspxphen, a metalloantibiotic whose influx seems to rely on hydrophobic pathways, independent of porins [39], as following presented.

#### 3.2.3. Computational Experiments

Computational approaches are usually focused on several amino acid residues characteristic of the constriction zone of OmpF channel. The constriction region of the porins has direct implications in the permeation of molecules, working as an exclusion region of the channel. The constriction area of OmpF is enriched with acidic amino acids, facing basic residues on the neighbouring wall, which contributes to the creation of a strong transverse electric field [26]. Based on these insights, the permeation of a molecule through porin channels strongly depends on its ability to interact with the constriction zone residues. OmpF-like porins commonly permeate molecules with sizes up to 600 Da [25], which may difficult the transport of metalloantibiotics, whose size varies between 700 and 800 Da [30].

The computational experiments carried out by Sousa et al. showed a reduced binding affinity of the metalloantibiotics to the constriction zone residues (Table 3), describing a more negative energy docking solution for residues located in the upper part of the porin (outside the constriction region) [39]. These results reveal a stronger interaction of the metalloantibiotics outside the constriction area, which hampers their transport through the channel. Mahendran et al. also previously anticipated that a stronger interaction with residues outside the constriction zone reduces the ability of a molecule to permeate across the channel [105]. These results corroborate the hydrophobic pathway previously proposed for metalloantibiotics, by biophysical and microbiological approaches, and agree with the assumption that metalloantibiotics may have a higher size than the limit able to cross the porin channel.

Combining the results obtained by distinct techniques, it is possible to infer that the influx of metalloantibiotics should privilege the diffusion across the lipid bilayer (being able to use the lipid/porin interface), being able to overcome resistance mechanisms based on the decreased permeation through porins. Moreover, these studies reinforce the relevance of employing multidisciplinary approaches for the understanding of drug influx pathways.

## 4. Antimicrobial Activity, Mechanism of Action, and Safety Profile of Metalloantibiotics

### 4.1. Antimicrobial Activity

The stability of the metalloantibiotics, together with the proposed alternative influx pathway in bacteria, triggered the study of the antimicrobial activity of these complexes. During the last years, the MICs of the metalloantibiotics have been assessed against bacterial control strains (*E. coli* ATCC 25922, *P. aeruginosa* ATCC 27853, *S. aureus* ATCC 25923 and *S. aureus* ATCC 29213) [30,33,34,35,36] (Table 4) and, more recently, the study of the antimicrobial activity was extended to multidrug-resistant (MDR) strains [36] (Table 5), chosen according to the first list of antibiotic-resistant “priority pathogens”, published by the World Health Organization (WHO) in 2017 [13]. This list highlights the main pathogens for which new antimicrobial drugs are promptly needed, attributing distinct priority stages to different bacteria. Among several pathogens, *E. coli*, *P. aeruginosa* (carbapenem-resistant), and *S. aureus* (methicillin-resistant) incorporate this list, with critical (Gram-negative strains) and high (Gram-positive strains) priority levels. These three clinically relevant bacterial strains exhibit multifaceted resistance mechanisms responsible for raising resistance to one or several antibiotics, including FQs [14,108,109,110,111,112,113,114].

The microbiological results reported for metalloantibiotics against control and clinical strains disclosed similar antimicrobial activity against susceptible strains and improved activity against MDR clinical strains (compared to free FQs), with greater activity against *S. aureus*. The microbiological assays performed against susceptible strains revealed similar MIC values for free FQs and metalloantibiotics, revealing the antimicrobial activity of these complexes. Moreover, the reduced MIC values described for metalloantibiotics in comparison to the ones determined for phen, Cu(II)/phen (1:1), and copper solutions (more than 10^3^-fold higher) reinforced the stability of the complexes, proving no dissociation [30,36], as previously explained. These results were also confirmed against all MDR clinical isolates. Concerning MDR strains, the MIC values disclosed pronounced differences between Gram-negative and Gram-positive clinical isolates. Regarding MDR *E. coli* and *P. aeruginosa* isolates, the MICs of the metalloantibiotics were analogous to the ones of free FQs, with the exception of Cucpxphen, which exhibited improved activity (4-fold) against two clinical isolates of *E. coli* (HSJ Ec002 and HSJ Ec003), out of the four tested. Moreover, metalloantibiotics did not reveal an advantage over FQs against *P. aeruginosa* isolates. The results stated by Ferreira et al. disclosed that metalloantibiotics alone are not able to bypass the bacterial resistance mechanisms to FQs of the studied clinical isolates [36]. However, metalloantibiotics revealed significant improved antibacterial activity (4 to 28-fold) against 15 out of 18 MRSA studied clinical isolates. Cucpxphen and Cuspxphen proved to be the metalloantibiotics with greater effectiveness against a higher number of isolates [36].

Drug combinations of antimicrobial agents belonging to different antibiotic classes is an approach broadly used in the clinical practice to fight antibiotic-resistant bacterial infections. FQs are commonly combined with β-lactams (as cephalosporins and/or carbapenems) or antifungal agents due to their known synergistic effect [115,116,117,118,119,120]. Although FQs do not have antifungal activity, their ability to bind to topoisomerase of fungi may explain the synergistic activity resulting from their combination with some antifungal agents [119]. Therefore, Ferreira et al. also tested some combinations of metalloantibiotics with other antimicrobial agents. Once again, differences against Gram-negative and Gram-positive clinical isolates were observed. The preliminary experiments revealed no synergistic activity of the metalloantibiotics against MDR *E. coli* isolates but uncovered synergistic or additive interactions against two MRSA isolates. These results urge a deeper research on the combination of metalloantibiotics with different classes of antimicrobial agents against distinct MDR clinical isolates. As previously pointed out, the combination of drugs usually adopts antimicrobial agents with different mechanisms of action. For this reason, the elucidation of the mechanism of action of metalloantibiotics was also clarified by the authors, as following presented.

### 4.2. Mechanism of Action

The mechanism of action of metalloantibiotics was recently clarified [36], although it was predicted to be comparable to the one of free FQs [30,121]. Therefore, metalloantibiotics were expected to intercalate into bacterial DNA, exhibiting nuclease activity [30,60]. Ferreira et al. carried out enzymatic inhibitory assays of DNA gyrases II and topoisomerases IV of *E. coli* and *S. aureus* and evaluated the effect of metalloantibiotics on bacterial membranes of the same strains by Atomic Force Microscopy (AFM) [36]. The experiments encompassed Cucpxphen and Cuspspxhen (and free cpx and spx as control) and confirmed the inhibitory enzymatic activity of the metalloantibiotics against both enzymes of Gram-negative and Gram-positive strains. Moreover, metalloantibiotics were revealed to be more effective against DNA gyrase of *E. coli* and topoisomerases IV of *E. coli* and *S. aureus.* The reduced susceptibility of the DNA gyrase of *S. aureus* to the action of metalloantibiotics was anticipated according to previous reports by some authors, pointing to the DNA gyrase as the preferential target of FQs in Gram-negative bacteria and the topoisomerase IV in Gram-positive bacteria [7,8,15]. The activity of the metalloantibiotics was comparable to the one of free cpx, with exception for topoisomerase IV of *S. aureus*, against which it was greater. This evidence supports the improved antimicrobial activity previously disclosed for metalloantibiotics against MRSA isolates [36].

In parallel, AFM experiments were carried out, presuming that metalloantibiotics may affect/damage bacterial membranes, due to the strong interaction previously reported with the lipidic component. This microscopic technique revealed no damage of the bacterial membranes in the presence of metalloantibiotics, corroborating an intracellular mechanism of action [36]. However, *E. coli* cells evidenced an increase in the cell size and filamentation upon treatment with metalloantibiotics and free FQs (Figure 6), a phenomenon previously reported by Diver et al. and Silva et al. after treatment with cpx [122,123]. The authors also described “ghost cells” and bacterial cells with a collapsed appearance upon treatment with all tested compounds, which is in agreement with previous studies of Cushnie et al. [124]. Concerning *S. aureus* cells, the authors observed a slight increase in cell size, reporting no significant changes upon cell treatment.

Hence, AFM experiments point to the intracellular mechanism of action of the metalloantibiotics, like the one of free FQs, as previously revealed by inhibitory enzymatic assays. Once again, the combination of enzymatic assays and microscopic techniques unraveled the importance of adopting distinct approaches to perform effective characterization of the mechanism of action of metalloantibiotics.

### 4.3. Safety Profile

The assessment of the safety profile is a mandatory step in the characterization of any compound intended to be used in the clinical practice. The in vitro toxicity of these antimicrobial drugs as FQs is commonly assessed by phototoxicity, cytotoxicity, and haemolysis assays [125,126,127], the combination of both in vitro and in vivo assays being extremely important. Concerning metalloantibiotics, toxicological safety data is scarce, only describing that these antimicrobial compounds are non-toxic against human cells (cytotoxicity 10 times higher than the MIC values determined against ATCC control strains), as stated by Sousa et al. [39] and Ferreira et al. [37]. Additionally, preliminary haemolysis assays disclosed no hemolytic activity of metalloantibiotics against human red blood cells (data of our research group, not published).

Moreover, tendinitis and tendon rupture (especially the Achilles tendon), related to the administration of FQs, is also commonly reported [128,129,130]. Thus, the risk of metalloantibiotics-induced tendon rupture can also be studied.

Therefore, extensive analysis of the safety profile of the metalloantibiotics is urgently needed.

## 5. Conclusions

Metalloantibiotics of Cu(II), FQs, and phen consist of a strategy widely explored to combat bacterial infections. These ternary complexes have been the subject of study over the last decades their stability being proved under physiological concentrations. In this work, we compiled the outcomes disclosed through the combination of potentiometric, spectroscopic, microscopic, microbiological, and computational approaches. Several works focused on the characterization of metalloantibiotics disclosed their influx pathways in bacterial cells, proved their antimicrobial activity against susceptible and clinical isolates of different bacterial strains, and clarified their mechanism of action. This review reinforces the potential of these metalloantibiotics as promising alternatives to free FQs.

We believe that some unexplored fields should also be investigated—the in-depth evaluation of the toxicity profile of metalloantibiotics, the assessment of antibiofilm activity and synergistic activity (involving several drug combinations), and the evaluation of their ability to circumvent resistant bacterial mechanisms based on efflux pumps.

## Figures and Tables

**Figure 1 microorganisms-09-01506-f001:**
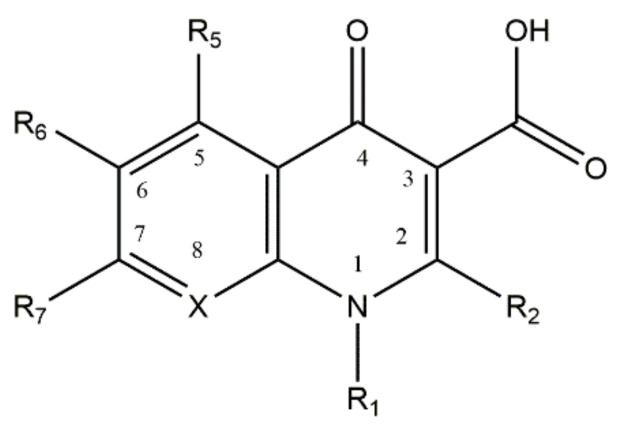
Structure of the base nucleus of quinolones and naphthyridones, drawn using ChemDraw Professional 17.0. The position 8 is occupied by a carbon or a nitrogen atom (substituted or non-substituted), in the case of quinolones or naphthyridones, respectively.

**Figure 2 microorganisms-09-01506-f002:**
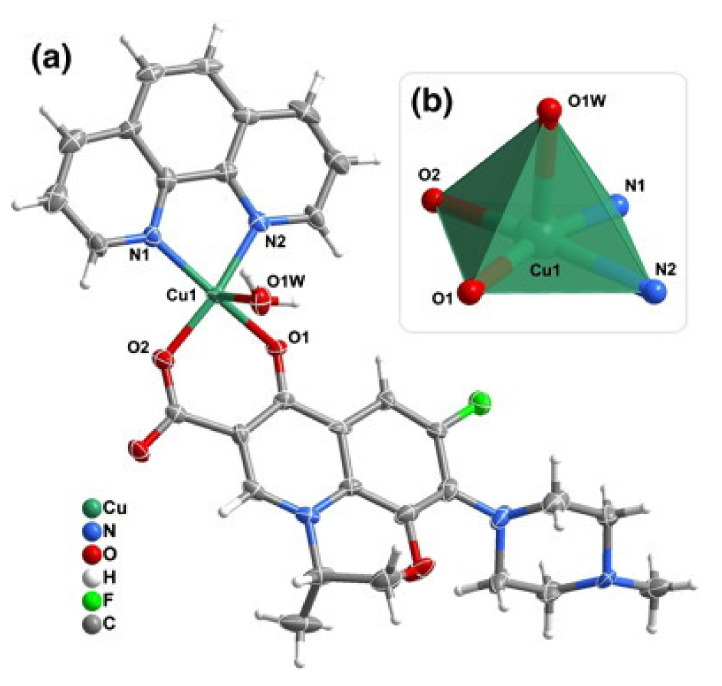
Representation of (**a**) the cationic complex of levofloxacin (lvx), [Cu(lvx)(phen)(H_2_O)]^+^, and of (**b**) the copper(II) five coordinated centre [CuN_2_O_3_] characteristic of the square pyramidal geometry slightly distorted of CuFQphen complexes. Reprinted from Journal of Inorganic Biochemistry, 110, Sousa, I., Claro, V., Pereira, J. L., Amaral, A. L., Cunha-Silva, L., de Castro, B., Feio, M. J., Pereira, E. and Gameiro, P., Synthesis, characterization and antibacterial studies of a copper (II) levofloxacin ternary complex., 64–71, Copyright 2012, with permission from Elsevier.

**Figure 3 microorganisms-09-01506-f003:**
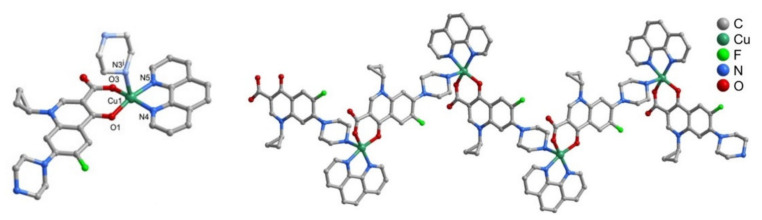
Representation of (**left**) the cationic complex [Cu(cpx)(phen)]^+^ and of (**right**) the coordination chain of [Cu(cpx)(phen)]^+^, formed in the crystal structure of Cucpxphen. Reprinted from Journal of Inorganic Biochemistry, 134, Feio, M. J., Sousa, I., Ferreira, M., Cunha-Silva, L., Saraiva, R. G., Queirós, C., Alexandre, J. G., Claro, V., Mendes, A., Ortiz, R., Lopes, S., Amaral, A. L., Lino, J., Fernandes, P., Silva, A. J., Moutinho, L., de Castro, B., Pereira, E., Perelló, L. and Gameiro, P., Fluoroquinolone–metal complexes: A route to counteract bacterial resistance?, 129–143, Copyright 2014, with permission from Elsevier.

**Figure 4 microorganisms-09-01506-f004:**
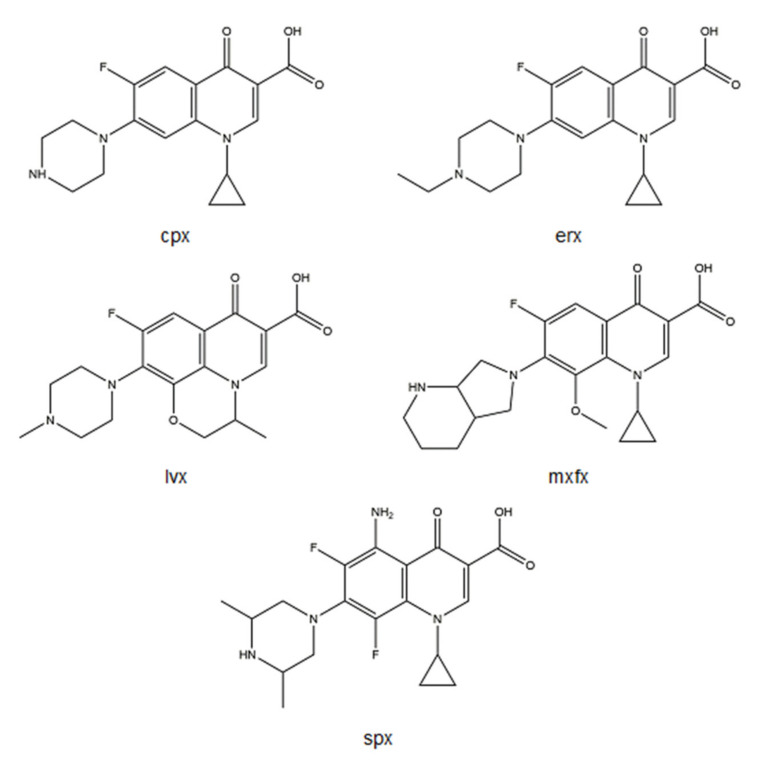
Structures of cpx, erx, lvx, mxfx, and spx, drawn using ChemDraw Professional 17.0.

**Figure 5 microorganisms-09-01506-f005:**
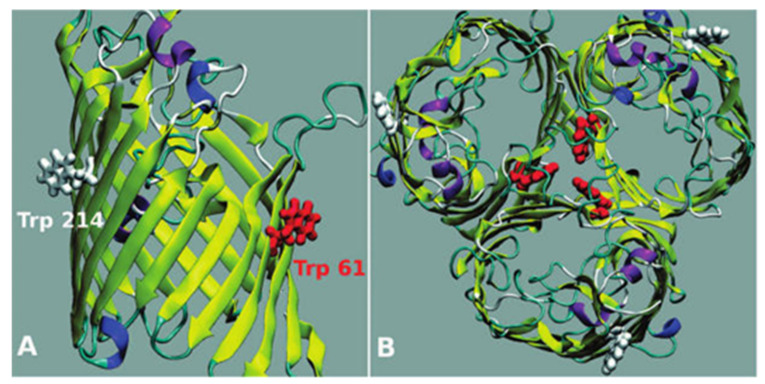
OmpF porin: a monomer side view (**A**) and the homotrimer top view (**B**). The two Trp residues of the monomers of the porin are represented in red (W61) and in white (W 214). Reprinted (adapted) with permission from Journal of the American Chemical Society, 130, Mach, T., Neves, P., Spiga, E., Weingart, H., Winterhalter, M., Ruggerone, P., Ceccarelli, M. and Gameiro, P., Facilitated Permeation of Antibiotics across Membrane Channels—Interaction of the Quinolone Moxifloxacin with the OmpF Channel, 13301-13309. Copyright 2008 American Chemical Society.

**Figure 6 microorganisms-09-01506-f006:**
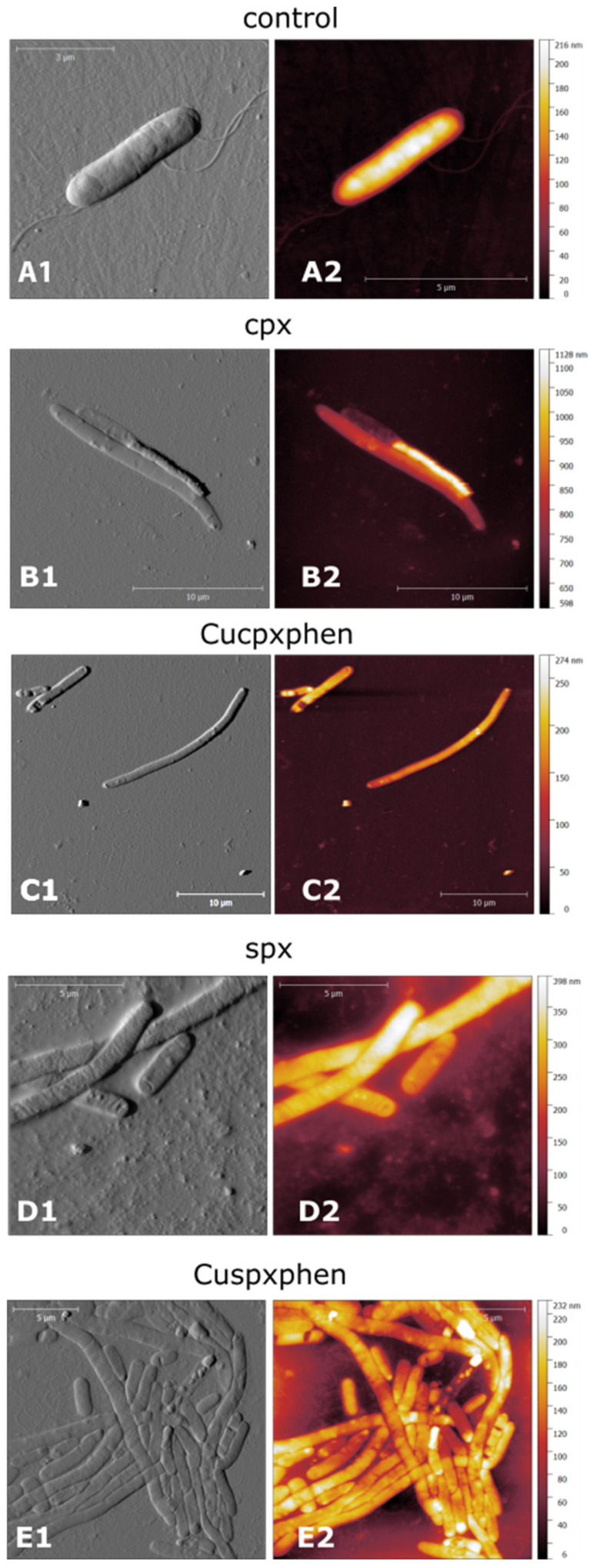
AFM images of *E. coli* ATCC 25922 control cells (**A**) and cells treated with cpx (**B**), Cucpxphen (**C**), spx (**D**), and Cuspxphen (**E**). A1, B1, C1, D1, and E1 are amplitude images; A2, B2, C2, D2, and E2 are height images. Images are representative of multiple areas from at least three samples analysed for each condition tested. Reprinted with permission from International Journal of Environmental Research and Public Health, 17, Ferreira, M., Bessa, L. J., Sousa, C. F., Eaton, P., Bongiorno, D., Stefani, S., Campanile, F. and Gameiro, P., Fluoroquinolone Metalloantibiotics: A Promising Approach against Methicillin-Resistant *Staphylococcus aureus*, 3127. Copyright 2020 MDPI.

**Table 1 microorganisms-09-01506-t001:** Stability constants of Cu(II) complexes (determined at 25 °C, I = 0.1 mol dm^−3^ NaCl) ± error majorant, from [30,33,34,35,51].

FQ	MHL Log β_1100_	ML Log β_11–10_	M(HL)_2_ Log β_1200_	ML_2_ Log β_12–20_	M(HL)A Log β_1101_	MLA Log β_11–11_
cpx [30]	5.48 ± 0.03	−2.49 ± 0.05	−	−6.13 ± 0.04	17.96 ± 0.05	8.42 ± 0.04
erx [33]	6.39 ± 0.03	−0.42 ± 0.04	10.97 ± 0.04	−3.32 ± 0.06	16.54 ± 0.05	9.21 ± 0.04
lvx [34]	6.22 ± 0.04	−1.02 ± 0.08	10.92 ± 0.05	−4.52 ± 0.07	16.53 ± 0.04	9.30 ± 0.09
mxfx [30]	6.01 ± 0.04	−1.32 ± 0.08	12.24 ± 0.07	−4.89 ± 0.04	15.95 ± 0.04	7.84 ± 0.09
spx [30]	6.09 ± 0.08	−	12.58 ± 0.08	−	16.46 ± 0.09	8.93 ± 0.09
ofx [35]	6.24 ± 0.04	−	11.20 ± 0.04	−	16.69 ± 0.03	9.21 ± 0.04
nfx [35]	6.95 ± 0.03	−	12.70 ± 0.04	−	17.56 ± 0.05	9.55 ± 0.04
lmx [51]	6.04 ± 0.01	−0.98 ± 0.03	10.51 ± 0.15	−5.48 ± 0.15	17.35 ± 0.07	8.45 ± 0.05

M is metal–Cu(II); HL is the FQ in the zwitterionic form; H is the proton; and A is phen. log β_1100_: M + HL⇌MHL; log β_11–10_: M + HL⇌ML + H; log β_1200_: M + 2HL⇌M(HL)_2_; log β_11–20_: M + 2HL⇌ML_2_ + 2H; log β_1101_: M + HL + A⇌MHLA; log β_11–11_: M + HL + A⇌MLA + H.

**Table 2 microorganisms-09-01506-t002:** Values for the partition constants ($K_{p}$ ± SD) of metalloantibiotics in different membrane mimetic systems of prokaryotic and eukaryotic cells, obtained by fluorescence and UV-vis spectroscopies and molecular dynamics (MD) simulations, from [38,71,75,77,87]. Speciation under physiological conditions (pH 7.4; concentrations in the same range of those used in the partition studies).

Metalloantibiotic and its Speciation	Membrane Mimetic System	Log 𝑲𝒑	Technique
Cucpxphen 97% di-cationic; 3% mono-cationic	DMPC	4.15 ± 0.11 [75]	Fluorescence ST
	POPC	3.2 ± 0.2 [38]	Fluorescence ST
	POPG	3.38 ± 0.09 [38]	Fluorescence ST
	POPG	3.69 ± 0.05 [38]	UV-vis
	POPC	−0.25–1.18 [38]	MD simulations
	POPG	1.99–2.47 [38]	MD simulations
	POPE:POPG (75:25)	4.94 ± 0.02 [75]	Fluorescence ST
	POPE:POPG (75:25)	4.45 ± 0.02 [75]	Fluorescence TR
	POPE:POPG:CL (67:23:10)	4.50 ± 0.03 [75]	Fluorescence ST
	POPE:POPG:CL (67:23:10)	4.15 ± 0.01 [75]	Fluorescence TR
	*E. coli* total lipid extract	5.32 ± 0.09 [75]	Fluorescence ST
	*E. coli* total lipid extract	4.65 ± 0.03 [75]	Fluorescence TR
Cuerxphen 42.7% di-cationic; 50.1% mono-cationic	DMPC	3.40 ± 0.03 ^a^	Fluorescence ST
	POPE:POPG (75:25)	4.49 ± 0.02 [87]	Fluorescence ST
	POPE:POPG:CL (67:23:10)	4.67 ± 0.01 [87]	Fluorescence ST
	*E. coli* polar lipid extract	5.18 ± 0.01 [87]	Fluorescence ST
	*E. coli* total lipid extract	4.85 ± 0.02 [87]	Fluorescence ST
Culvxphen 38.5% di-cationic; 57% mono-cationic	DMPC	4.36 ± 0.03 ^a^	Fluorescence ST
	POPE:POPG:CL (67:23:10)	4.33 ± 0.10 ^a^	Fluorescence ST
	*E. coli* total lipid extract	4.26 ± 0.04 ^a^	Fluorescence ST
Cumxfxphen 82.7% di-cationic; 16.1% mono-cationic	DMPC	N.d. [77]	Fluorescence ST
	POPE:POPG (75:25)	4.22 ± 0.10 [77]	Fluorescence ST
	POPE:POPG:CL (67:23:10)	4.38 ± 0.01 [77]	Fluorescence ST
	*E. coli* polar lipid extract	4.39 ± 0.02 [77]	Fluorescence ST
	*E. coli* total lipid extract	4.75 ± 0.01 [77]	Fluorescence ST
Cuspxphen 52.6% di-cationic; 38.1% mono-cationic	POPE:POPG (75:25)	5.05 ± 0.08 [71]	Fluorescence ST
	POPE:POPG (75:25)	5.04 ± 0.04 [71]	Fluorescence TR
	*E. coli* total lipid extract	4.87 ± 0.08 [71]	Fluorescence ST
	*E. coli* total lipid extract	5.24 ± 0.01 [71]	Fluorescence TR

$K_{p}$ values calculated taking into account the water molar concentration (55.3 mol dm^−3^, at 37 ˚C [78]). ^a^ Data determined by Gameiro’s research group in liposomes prepared in HEPES buffer (10 mmol dm^−3^, pH 7.4, NaCl 0.1 mol dm^−3^). Data not published. N.d.-not possible to determine with the experimental data obtained; ST means Steady-State; TR means Time-Resolved.

**Table 3 microorganisms-09-01506-t003:** ΔΔG _binding_ for the docking of Cucpxphen and Cuspxphen with the OmpF protein using a docking box adjusted to the constriction zone of the channel. Reproduced from [39] with permission from the Royal Society of Chemistry.

Metalloantibiotic	ΔΔG _binding_/Kcal Mol^−1^			
	Rigid Docking Protocol		Flexible Docking Protocol	
	4KRA	2OMF	4KRA	2OMF
Cucpxphen	1.2	−0.01	2.6	2.6
Cuspxphen	1.0	0.3	2.9	3.1

Rigid and flexible docking protocols were used and two protein models, 2OMF and 4KRA, were considered.

**Table 4 microorganisms-09-01506-t004:** Minimum inhibitory concentration (MIC) values of several metalloantibiotics (CuFQphen) against reference strains *E. coli* ATCC 25922, *P. aeruginosa* ATCC 27853, *S. aureus* ATCC 25923, and *S. aureus* ATCC 29213, from [30,33,34,35,36].

Metalloantibiotic	MIC Value/μmol dm^−3^ (μg mL^−1^)			
	*E. coli* ATCC 25922	*P. aeruginosa* ATCC 27853 [36]	*S. aureus* ATCC 25923 [36]	*S. aureus* ATCC 29213 [36]
Cucpxphen	0.011 [36]–0.026 [30] (0.008–0.016)	0.17–0.35 (0.12–0.25)	0.35–0.71 (0.25–0.5)	1.41 (1)
Cuerxphen	0.022 [36]–0.027 [33] (0.015)	2.97 (2)	0.37–0.74 (0.25–0.5)	0.18–0.37 (0.12–0.25)
Culvxphen	0.021 [36]–0.07 [34] (0.015–0.030)	1.39 (1)	0.35–0.69 (0.25–0.5)	0.35 (0.25)
Cumxfxphen	0.018 [30]–0.038 [36] (0.015–0.030)	2.54 (2)	0.08–0.15 (0.06–0.12)	0.08–0.15 (0.06–0.12)
Cuspxphen	0.005 [36]–0.020 [30] (0.004–0.016)	0.65 (0.5)	0.32 (0.25)	0.08–0.16 (0.06–0.12)
Cuofxphen	0.050 [35]–0.083 [30] (0.03)	−	−	−
Cunfxphen	0.107 [35]–0.185 [30] (0.06)	−	−	−

**Table 5 microorganisms-09-01506-t005:** Minimum inhibitory concentration (MIC) values of several free FQs and metalloantibiotics (CuFQphen) against multidrug-resistant (MDR) strains of *E. coli* (HSJ Ec002 and HSJ Ec003) and MRSA (Sa1-SA3, Sa3-SA3, 19/35 and 26/01), from [36].

Compound	MIC Value/μmol dm^−3^ (μg mL^−1^)					
	*E. coli* HSJ Ec002	*E. coli* HSJ Ec003	MRSA Sa1-SA3	MRSA Sa3-SA3	MRSA 19/35	MRSA 26/01
cpx	386.3 (128)	193.2 (64)	386.3 (128)	386.3–772.6 (128–256)	≥3090.5 (≥1024)	1545.2 (512)
Cucpxphen	93.0 (64)	45.1–90.3 (32–64)	90.3 (64)	90.3 (64)	180.5 (128)	180.5 (128)
erx	178.1 (64)	178.1 (64)	22.3 (8)	44.5–89.0 (16–32)	178.1 (64)	712.3 (256)
Cuerxphen	95.0 (64)	95.0 (64)	95.0 (64)	95.0 (64)	95.0 (64)	95.0 (64)
lvx	22.1 (8)	88.6 (32)	708.4 (256)	44.3 (16)	177.1 (64)	1416.8 (512)
Culvxphen	44.4 (32)	44.4–88.7 (32–64)	88.7 (64)	88.7–177.5 (64–128)	88.7 (64)	88.7 (64)
mxfx	18.3 (8)	18.3 (8)	18.3 (8)	292.3 (128)	18.3 (8)	73.1 (32)
Cumxfxphen	20.3 (16)	40.6 (32)	10.2 (8)	10.2 (8)	20.3 (16)	40.6 (32)
spx	40.8 (16)	81.6 (32)	652.4 (256)	20.4 (8)	163.1 (64)	652.4 (256)
Cuspxphen	41.6 (32)	83.1 (64)	83.1–166.2 (64–128)	41.6 (32)	41.6 (32)	83.1 (64)

MDR strains presented revealed the most relevant results (improved antimicrobial activity of metalloantibiotics compared to free FQs) among several studied strains.

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
