# Peer review of "Fluoroquinolone-Transition Metal Complexes: A Strategy to Overcome Bacterial Resistance"

_microorganisms, 2021, doi:10.3390/microorganisms9071506_

Round 1

Reviewer 1 Report

In this paper, Ferreira and Gameiro review the role of fluoroquinolone-metal complexes as agents to overcome bacterial resistance to antibiotics. I have a few suggestions for the authors, which I hope will help improve the clarity of their manuscript:

  1. In section 2, the authors mention that FQs have multiple ionisable sites and hence different charge states depending on the pH, but do not expand on the role this plays in drug permeation across cellular membranes. It would help the manuscript if this aspect were mentioned briefly – the molecules will have a different charged state depending on the pH, which can dramatically influence membrane permeability (see Nikaido and Thanassi, AAC 1993; Cama et al Sci. Reps. 2016). This is clinically relevant for FQs and may be so for the metalloantibiotics discussed here – for eg., E. coli (which the authors discuss in detail later) is the causative agent in the majority of urinary tract infections, and the pH of urine is known to vary in the range of 4.5 – 8. So just looking at pH 7.4 as “physiological” is not enough when developing new antibacterial agents.
  2. In section 3.1, there is much discussion about partition constants, but there are better ways to quantify membrane permeability in model systems, such as the liposomes briefly mentioned – the authors should mention other techniques here, such as the PAMPA assay which remains popular. Further, the type of lipid can influence permeability as well, another reason why lipid based models are important (see Purushothaman et al Soft Matter 2016 for eg).
  3. In table 2 please specify whether the errors are s.d. or s.e.m.
  4. Line 244 – there is discussion about the enhanced partition constants of some of the complexes, have any of these results been translated into observed enhanced accumulation in cells? If so please give examples, and if not, this could be suggested as future work (there are lots of new techniques for studying the cellular accumulation of antibiotics that should be mentioned, particularly since a number of them study FQs – see for eg. Vergalli et al. Sci. Reps. 2017, Cama et al. Lab Chip 2020 for fluorescence based techniques, Prochnow et al. Analytical Chem. 2019 for a Mass Spec protocol).
  5. Lines 260-263 – it would help if there were more discussion about the double membrane permeability barrier of gram-negatives. While correct that FQs usually get across the outer membrane via porins, they still have to diffuse across the inner cytoplasmic lipid barrier – so it is a combination of routes, not simply a purely hydrophobic or hydrophilic route. The vesicle and other studies mentioned above show clearly that FQs can also diffuse across pure lipid membranes very easily (depending on their charge state).
  6. I found lines 290-295 confusing, because initially it is suggested that the metalloantibiotics translocate independently of porins, whereas later in the paragraph there are counter examples provided. This should be rewritten more clearly, and the initial statement that the route of transport is “porin independent” should be made less strong, since there is a follow up counter example provided.
  7. Line 375 – multidisciplinary approaches are mentioned, but I reiterate the point that cellular accumulation studies are likely to be critical, and these are not really discussed at all in the manuscript. Strongly suggest that these are included, even as potential future work on these new antibiotics – since both fluorescence and mass spec techniques are likely to be useful here.
  8. Tables 4 and 5 – I strongly suggest the authors consider reporting their values in ug/ml rather than molar concentrations, so that easy comparison with standard MIC reporting (CLSI breakpoints etc) is facilitated. The clinical relevance of the enhanced activity of the metalloantibiotics for the MDR strains is not immediately obvious with the data as currently presented.
  9. In the discussion (line 400 onwards) about the role of the metalloantibiotics in combatting MDR strains – does this not depend on the resistance mechanism involved? So do the authors know whether these new agents can overcome, say, reduced influx associated resistance but not target-mutation derived resistance, for example? Some clarification required here.
  10. Line 424 – why would you use an anti-fungal on a bacterial infection?
  11. Fig. 6 – more explanation of what the figure means is required in the caption. What do the differences in height mean, for instance? Please explain the purpose of the image in detail in the caption.
  12. In the short para (4.3) on safety profiles, there is no mention of one of the biggest problems with FQ treatment, i.e. increased propensity for tendon rupture in patients. This should be mentioned, and specifically that these new agents should also be checked for this toxic effect.
  13. Some of the figures are quite pixelated, would help to have higher resolution images provided.

Author Response

We thank the reviewer for the comments and have performed the recommended changes, as following described.

Point 1: “In section 2, the authors mention that FQs have multiple ionisable sites and hence different charge states depending on the pH, but do not expand on the role this plays in drug permeation across cellular membranes. It would help the manuscript if this aspect were mentioned briefly – the molecules will have a different charged state depending on the pH, which can dramatically influence membrane permeability (see Nikaido and Thanassi, AAC 1993; Cama et al Sci. Reps. 2016). This is clinically relevant for FQs and may be so for the metalloantibiotics discussed here – for eg., E. coli (which the authors discuss in detail later) is the causative agent in the majority of urinary tract infections, and the pH of urine is known to vary in the range of 4.5 – 8. So just looking at pH 7.4 as “physiological” is not enough when developing new antibacterial agents.”

Authors’ response: The presence of different species among the pH range and its implication in the interaction/penetration across the membrane were mentioned in this section. The suggested references were also added to the manuscript.

Point 2: “In section 3.1, there is much discussion about partition constants, but there are better ways to quantify membrane permeability in model systems, such as the liposomes briefly mentioned – the authors should mention other techniques here, such as the PAMPA assay which remains popular. Further, the type of lipid can influence permeability as well, another reason why lipid based models are important (see Purushothaman et al Soft Matter 2016 for eg).”

Authors’ response: We added some proposals for future experiments as PAMPA assay and membrane permeabilization assays using carboxyfluorescein and SYTOX Green.

Point 3: “In table 2 please specify whether the errors are s.d. or s.e.m..”

Authors’ response: We performed the specification of the error of the values of Tables 1 and 2.

Point 4: “Line 244 – there is discussion about the enhanced partition constants of some of the complexes, have any of these results been translated into observed enhanced accumulation in cells? If so please give examples, and if not, this could be suggested as future work (there are lots of new techniques for studying the cellular accumulation of antibiotics that should be mentioned, particularly since a number of them study FQs – see for eg. Vergalli et al. Sci. Reps. 2017, Cama et al. Lab Chip 2020 for fluorescence based techniques, Prochnow et al. Analytical Chem. 2019 for a Mass Spec protocol).”

Authors’ response: Accumulation studies were not performed. We thank the comment and also added accumulation studies as future work, in parallel to membrane permeability assays proposed in Point 2.

Point 5: “Lines 260-263 – it would help if there were more discussion about the double membrane permeability barrier of gram-negatives. While correct that FQs usually get across the outer membrane via porins, they still have to diffuse across the inner cytoplasmic lipid barrier – so it is a combination of routes, not simply a purely hydrophobic or hydrophilic route. The vesicle and other studies mentioned above show clearly that FQs can also diffuse across pure lipid membranes very easily (depending on their charge state).”

Authors’ response: We thank the comment and have enlarged the discussion on the membrane permeability barrier of Gram-negative bacteria.

Point 6: “I found lines 290-295 confusing, because initially it is suggested that the metalloantibiotics translocate independently of porins, whereas later in the paragraph there are counter examples provided. This should be rewritten more clearly, and the initial statement that the route of transport is “porin independent” should be made less strong, since there is a follow up counter example provided.”

Authors’ response: We thank the comment but these paragraphs describe the microbiological experiments. In this case, the results clearly show that the influx of metalloantibiotics is independent on porins, with exception of the ones of cpx and lvx. For this reason, we did not perform the suggested change.

Point 7: “Line 375 – multidisciplinary approaches are mentioned, but I reiterate the point that cellular accumulation studies are likely to be critical, and these are not really discussed at all in the manuscript. Strongly suggest that these are included, even as potential future work on these new antibiotics – since both fluorescence and mass spec techniques are likely to be useful here.”

Authors’ response: We thank the comment and, as pointed in Point 4, accumulation studies were suggested in the manuscript as future work to carry out.

Point 8: “Tables 4 and 5 – I strongly suggest the authors consider reporting their values in ug/ml rather than molar concentrations, so that easy comparison with standard MIC reporting (CLSI breakpoints etc) is facilitated. The clinical relevance of the enhanced activity of the metalloantibiotics for the MDR strains is not immediately obvious with the data as currently presented.”

Authors’ response: We thank the comment and added the values of the MICs in ug/mL.

Point 9: “In the discussion (line 400 onwards) about the role of the metalloantibiotics in combatting MDR strains – does this not depend on the resistance mechanism involved? So do the authors know whether these new agents can overcome, say, reduced influx associated resistance but not target-mutation derived resistance, for example? Some clarification required here.”

Authors’ response: The MDR clinical isolates used in the referred study were acquired from hospitals. Although the resistance profiles are known (through the disk diffusion method), the bacterial mechanisms of resistance were not studied. Among the several studied clinical isolates, some of them may share the same mechanisms of resistance but it stills unknown.

Point 10: “Line 424 – why would you use an anti-fungal on a bacterial infection?”

Authors’ response: Although FQs do not have antifungal activity, their ability to bind to the topoisomerase of fungi may explain the synergistic activity resultant from their combination with some antifungal agents (10.1093/jac/dkn473). The sentence and reference were added to the manuscript.

Point 11: “Fig. 6 – more explanation of what the figure means is required in the caption. What do the differences in height mean, for instance? Please explain the purpose of the image in detail in the caption.”

Authors’ response: The purpose of the figure is explained in page 14 line 543, We don’t understand what the reviewer means with this comment.

Point 12: “In the short para (4.3) on safety profiles, there is no mention of one of the biggest problems with FQ treatment, i.e. increased propensity for tendon rupture in patients. This should be mentioned, and specifically that these new agents should also be checked for this toxic effect.”

Authors’ response: We thank the comment and added the evaluation of this toxic effect to the mentioned section, as well as references.

Point 13: “Some of the figures are quite pixelated, would help to have higher resolution images provided.”

Authors’ response: Figure 1 was provided with higher resolution. The figures 2, 3 and 5 were obtained from previous papers. To obtain higher resolution pictures it is necessary more time to contact the sources.

Reviewer 2 Report

The authors present an interesting review on metalloantibiotics, focused on fluoroquinolones, with a new and significant contribution to this field. Publication is recommended.

Author Response

We thank the reviewer for the comments.

Reviewer 3 Report

Review article entitled "Fluoroquinolone-transition metal complexes: a strategy to overcome bacterial resistance" is focused on antibiotic resistance of fluoroquinolones (FQs) and their transition metal complex with FQs a possibility to overcome the resistance. Authors have also discussed about stability of FQ metalloantibiotics, possible translocation pathways  and related various tools and techniques.

Review comments:

The article is good written, however needs simplification of complex sentences having no conjunctions. The article is full of complex sentences for e.g.

"The chemical structure of quinolones, derived from a 4-oxo-1,8-naphthyridine-3-carboxylic acid nucleus (Figure 1), contains a conserved carboxylic acid and an exocyclic oxygen in the positions 3 and 4, respectively, responsible for the antimicrobial activity of this family, and a carbon atom (unsubstituted or substituted) in position 8 [1,2,4]" 

"The following decades yielded different generations of FQs, classified according to the spectrum of action, potency, pharmacodynamics and pharmacokinetics (bioavailability) properties [1,2]."

Authors have mentioned about only four metals (copper(II), iron(II), cobalt(II), nickel(II) and zinc(II) and focused on copper(II) complexes due copper(II) FQs are more stable. However, authors should also include experimental FQs in complexes with other transitions metals and their future possibility, for e.g. Please refer the research article "Vieira et al., Synthesis and antitubercular activity of palladium and platinum complexes with fluoroquinolones, Eur. J. Med. Chem, 2009, 44(10), 4107-4111"

Cited references list has typographical errors e.g. Refs 45, 53, 56, 58.

Reference 98 has no page numbers

Many references lack DOI numbers that needs to be added.

Author Response

We thank the reviewer for the comments and have performed the recommended changes, as following described. The changes were highlighted in the revised document. The reference list was carefully corrected.

Point 1: “The article is good written, however needs simplification of complex sentences having no conjunctions. The article is full of complex sentences for e.g. 

  1. "The chemical structure of quinolones, derived from a 4-oxo-1,8-naphthyridine-3-carboxylic acid nucleus (Figure 1), contains a conserved carboxylic acid and an exocyclic oxygen in the positions 3 and 4, respectively, responsible for the antimicrobial activity of this family, and a carbon atom (unsubstituted or substituted) in position 8 [1,2,4]" 

  1. "The following decades yielded different generations of FQs, classified according to the spectrum of action, potency, pharmacodynamics and pharmacokinetics (bioavailability) properties [1,2]."

Authors’ response: The complex sentences were changed according to the reviewer comment.

 Point 2: “Authors have mentioned about only four metals (copper(II), iron(II), cobalt(II), nickel(II) and zinc(II) and focused on copper(II) complexes due copper(II) FQs are more stable. However, authors should also include experimental FQs in complexes with other transitions metals and their future possibility, for e.g. Please refer the research article "Vieira et al., Synthesis and antitubercular activity of palladium and platinum complexes with fluoroquinolones, Eur. J. Med. Chem2009, 44(10), 4107-4111"

 Authors’ response: The suggested article was added to the manuscript and other transition metals were mentioned.

Point 3: “Cited references list has typographical errors e.g. Refs 45, 53, 56, 58. Reference 98 has no page numbers. Many references lack DOI numbers that needs to be added.”

Authors’ response: The authors thank the reviewer for the comments. The reference list was carefully corrected.

Round 2

Reviewer 3 Report

Revision of the review article entitled "Fluoroquinolone-transition metal complexes: a strategy to overcome bacterial resistance" is satisfactory and therefore I recommend to accept this article for publication.